# Ground Subsidence Monitoring in a Mining Area Based on Mountainous Time Function and EnKF Methods Using GPS Data

**Shifang Zhang and Jin Zhang \***

College of Mining Engineering, Taiyuan University of Technology, Taiyuan 030024, China
\* Correspondence: zhangjin@tyut.edu.cn

**Abstract:** Ground subsidence is an important geomorphological phenomenon in mining areas. It is difficult to monitor and predict ground subsidence with high precision, especially in mountainous mining areas. Taking the mining workface of a mountainous coalfield in Taiyuan City, in the Shanxi Province of China as an example, this research selects five typical points from GPS observation data along the strike section. Based on the materials, the ground subsidence processes at these typical points are monitored and predicted using the mountainous time function method. Acquired from the mountains time function is a recurrence equation, which is regarded as the state equation, and the Ensemble Kalman (EnKF) method is conducted accordingly. Finally, the performance of the two methods is evaluated and compared using error curves and indexes. This research presents a recurrence equation based on the mountainous time function method and establishes the EnKF method for ground subsidence monitoring and prediction. Meanwhile, compared to the mountainous time function method, the values of the *ME, MAE, RMSE* and *MAPE* indexes are largely improved for the EnKF method. Hence, this research not only presents an effective method for ground subsidence monitoring in mountainous mining areas, but also provides theoretical support for safe coal mining and environmental protection.

**Keywords:** ground subsidence monitoring and prediction; mountainous mining area; mountainous time function method; EnKF method; error index; GPS observation data

## 1. Introduction

As one of the most important fossil energy sources, coal plays an important role in supporting the world's economy. China is the world's largest coal producer [1], and about one-third of the total coalfields in China are located in mountainous areas [2]. Ground subsidence due to coal mining may lead to significant damage to buildings, roads, water facilities and other surface objects [3,4]. Moreover, although the mining hollows can be used as a tourist attraction, they also threaten ecological and social stability [5,6].

Ground subsidence in mining areas is a complicated four-dimensional spatiotemporal process [7–9]. To reduce the damage resulting from the ground subsidence process, many researchers have conducted relevant studies that focus on predicting subsidence [10]. Knothe presented a time function method to predict the subsidence process in relation to coal mining in 1953 [11]. Using the probability integral method, Hu et al. demonstrated the parameter computation method for the time function method [8,12]. Taherynia et al. combined the Knothe- and Geertsma-influenced functions to model the subsidence process in the South Pars Gas Field of Iran [13]. Polanin proposed a Knothe–Budryk method, which predicted the subsidence using two components of the influence function [14]. Zhang et al. developed the time function model by using two parameters for predicting the ground subsidence process of a mining area in Hezi City, China [3]. Cheng presented a developed time function model based on the inverse function of unstable creep [15]. Meanwhile, there are many other models for predicting the ground subsidence process in mining areas,

such as the logistic function, the arc tangent function, the negative exponential method, the Weibull time sequence function, the Bertalanffy time function and the normal time function [16–20]. However, most of these models are widely applied in flat mining areas or they assume the experimental areas to be flat regions.

Due to complex topographic and geologic conditions, it is much more difficult to predict the ground subsidence process in mountainous mining areas [2]. To resolve this issue, He and Kang studied the laws of ground subsidence in mountainous mining areas [21] and analyzed the observation data of ground subsidence in the Xishan Coalfield of Shanxi Province, China [22]. Guo et al. predicted the ground subsidence in mountainous areas based on Knothe's model [23]. Meanwhile, Lian et al. deduced a ground subsidence prediction model in mountainous mining area [2]. Compared to the traditional leveling surveying method, the developed technologies such as GPS and InSAR provide a significant opportunity for the in-depth study of the ground subsidence process in mountainous mining areas [1,4,24,25]. Regrettably, the models to predict the ground subsidence process seldom consider the observation data in the prediction process [3,26].

As one of the most widely applied methods in the field of data processing, the Kalman filter method can assimilate the model simulation results and the observation data in the prediction process [27]. Coupling an ensemble prediction and the Kalman filter, the ensemble Kalman filter (EnKF) utilizes ensemble methods to predict the state variables and the error covariance matrix [28,29]. With fewer computational burdens, the EnKF method does not require a tangent linear operator or adjoint model [30]. Due to these advantages, the EnKF method has seen success in many fields, such as remote sensing indexes [31–33], weather forecasting [34,35], tracking the spread of epidemics [36,37] and so on.

Therefore, this research aims to present a novel EnKF method in subsidence prediction by taking mountainous time function as the state equation [38]. By selecting typical observation points, the ground subsidence process is modeled and predicted using both the EnKF and the mountainous time function methods. Then, the performance of the two methods is evaluated and compared through different indexes. This research not only extends the fields in which the EnKF method can be applied, but also provides a reference for the sensible exploitation of coal resources and environmental protection [39–42].

The rest of this paper is organized as follows. Section 2 describes the study area and the materials, including the data source and its preliminary process. Section 3 presents the mountainous time function and the developed EnKF method. Section 4 analyzes the subsidence curves and error curves of the typical points using the two methods, and it computes the error indexes of the two methods. Section 5 discusses the results and previous research, and presents the innovations of this research and its future prospects. Section 6 offers a final conclusion.

## 2. Study Area and Materials

### 2.1. Study Area

The study area examined in this research is a workface of the Tunlan Minefield in Xishan Coalfield. The location of the study area is shown in Figure 1, as the red star in the upper right sub-figure. The Xishan Coalfield is located in the western part of Taiyuan City, the capital city of Shanxi Province. It is in the middle and northern part of China.

The topography of the Tunlan Minefield is complex. The main geomorphological types are structural erosion geomorphology and fluvial deposition geomorphology [22]. With a severe incision, the study area displays thick residual loess on the ridges and hills.

Based on the elevation distribution of the strike and incline sections, the workface is in an undulating topography with a series of ridges and valleys. The mountainous ground surface significantly increases the difficulty of predicting the ground subsidence as the coal resources exploit.

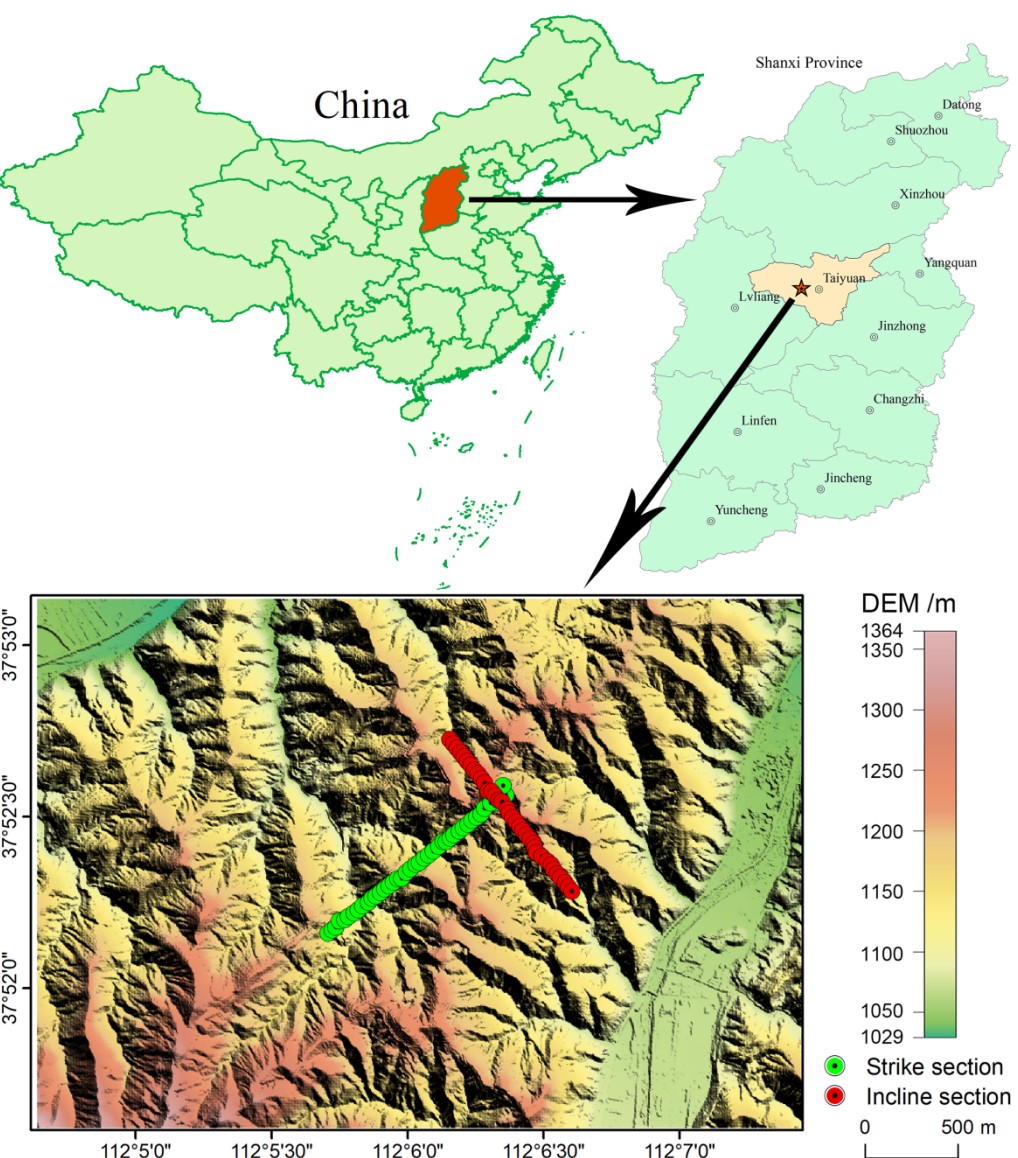

**Figure 1.** Study area and the distribution of the workface.

### 2.2. Materials

2.2.1. Study Area: Basic Information

Compared to areas characterized by flat topography, the ground subsidence process is difficult to predict in mountainous terrain due to its complex topographical and geological conditions [2]. To quantify this effect, He et al. presented the ground characteristic coefficient for the mining process in mountainous regions [43].

In this research, the ground characteristic coefficient is determined by basic information about the study area, including the slope, curvature, land use and geologic characteristics. The distribution of these characteristics is shown in Figure 2.

Figure 2 has the same geographic frame as Figure 1, so the geographic frame is omitted in Figure 2. In Figure 2, the slope and curvature are computed based on digital elevation model (DEM) data with a 5 m resolution. The land use and geology were acquired from thematic maps in vector format.

Figure 2 shows that the slope and curvature change largely along the strike section and incline section due to the data source. Along the strike section, there are no escarpments or very steep areas. As for the land use, the strike section is mainly distributed in forestland, and the incline section ranges from forestland to grassland and cropland in the NW–SW

direction. In addition, both the strike section and the incline section are located in the Shihezi and Shangya groups.

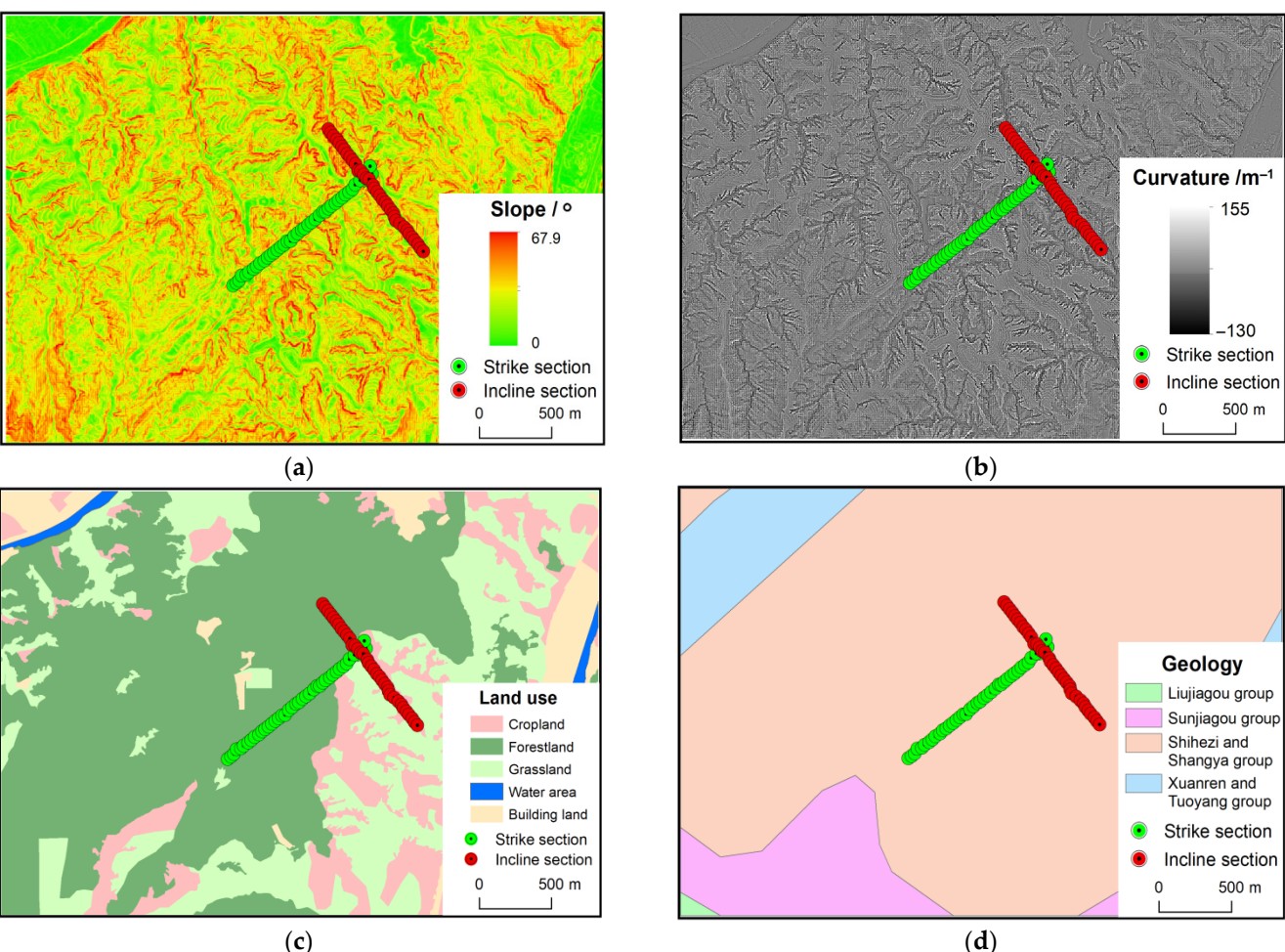

**Figure 2.** The characteristics of the study area: (**a**) slope; (**b**) curvature; (**c**) land use; (**d**) geology.

2.2.2. Workface: Basic Information

Basic information about the workface is shown in Table 1.

**Table 1.** The basic situation of the workface.

| Item | Value | Item | Value |
|---|---|---|---|
| Ground elevation/m | 1110–1241 | Incline section length/m | 235 |
| Workface elevation/m | 642–730 | Coal thickness/m | 1.8–3.15 |
| Strike section length/m | 1992–1999 | Coal inclination angle/° | 2–8 |

Table 1 presents the values of the ground elevation, workface elevation, strike section length, incline section length, coal thickness and coal inclination angle of the workface. These values provide the basis for predicting the ground subsidence process. The data also indicate that the workface is characterized by an undulating topography, which increases the difficulty of prediction.

In addition, the propulsion distance of the workface in each month is computed in Table 2.

From Table 2, the propulsion velocity can be acquired. This is an important parameter in ground subsidence modeling. Meanwhile, the mining process can be inferred from the propulsion distances of the workface in each month. Overall, the propulsion velocity is deemed to be constant in each period.

**Table 2.** The propulsion meter of the workface in each month.

| Month | Propulsion Distance/m | Month | Propulsion Distance/m |
|---|---|---|---|
| 2018.07 | 69.6 | 2019.07 | 73.6 |
| 2018.08 | 88.8 | 2019.08 | 99.2 |
| 2018.09 | 96.8 | 2019.09 | 77.6 |
| 2018.10 | 55.2 | 2019.10 | 81.6 |
| 2018.11 | 76.0 | 2019.11 | 88.0 |
| 2018.12 | 64.8 | 2019.12 | 75.2 |
| 2019.01 | 86.4 | 2020.01 | 64.8 |
| 2019.02 | 84.8 | 2020.02 | 91.2 |
| 2019.03 | 79.2 | 2020.03 | 82.4 |
| 2019.04 | 106.4 | 2020.04 | 61.6 |
| 2019.05 | 88.0 | 2020.05 | 41.6 |
| 2019.06 | 86.4 | | |

## 3. Methods

Figure 3 presents the computation process of this research. Primarily, it predicts the typical observation points along the strike section, so the typical observation points should be determined first. Then, the x-coordinates of these points are computed from their locations; the influence radius is acquired based on the mining depth and the influence angle, and the time influence parameter is achieved according to the advancing rate and critical mining dimension. Based on the three computed parameters, the time function method can be conducted accordingly. The ground characteristic coefficient can be determined based on slope, curvature, soil, geology and land use status. The mountainous time function method can be conducted based on the time function method, slope and ground characteristic coefficient for the typical observation points. Taking the mountainous time function as the state equation, the EnKF method is achieved through several steps using an iteration process. Finally, the performance of the mountainous time function and the EnKF methods is evaluated through subsidence curves, error curves and error indexes.

Based on the workflow, the typical observation points along the strike section are determined first; then, the mountainous time function method is presented, and the EnKF method is established by using the mountainous time function method as the state equation; finally, the error indexes are selected to quantify and estimate the prediction quality based on the two methods.

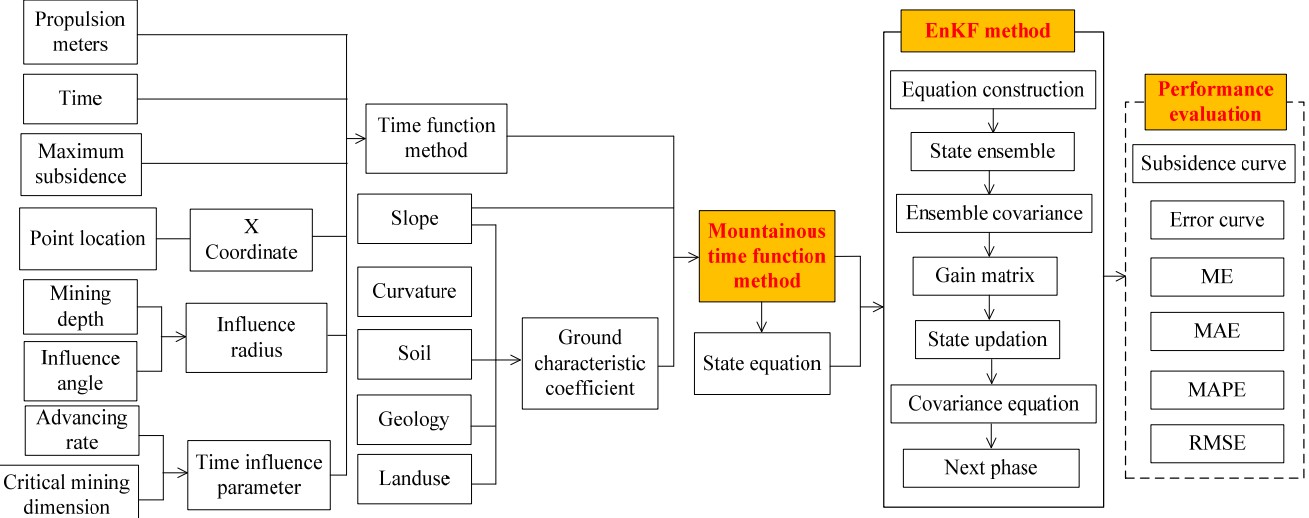

**Figure 3.** Work flow of this research.

### 3.1. Determination of Typical Observation Points

There are many observation points along the section. In this research, the subsidence process is modeled and predicted through typical observation points along the strike section. To determine the typical observation points, the x-coordinates of the points should be acquired first.

According to the open-off cut position and the offset distance (15 m in this research) of the workface, the x-coordinates of the observation points along the strike section were computed based on their locations and are shown in Table 3.

**Table 3.** The x-coordinates of the observation points along the strike section (m).

| Observation Point | x-Coordinate | Observation Point | x-Coordinate |
|---|---|---|---|
| A1 | −587.256 | A23 | 71.857 |
| A2 | −558.582 | A24 | 100.574 |
| A3 | −532.639 | A25 | 133.520 |
| A4 | −498.651 | A26 | 172.987 |
| A5 | −470.135 | A27 | 206.542 |
| A6 | −444.269 | A28 | 219.384 |
| A7 | −407.411 | A29 | 249.868 |
| A8 | −379.774 | A30 | 275.231 |
| A9 | −348.981 | A31 | 310.314 |
| A10 | −312.773 | A32 | 338.719 |
| A11 | −284.147 | A33 | 370.850 |
| A12 | −259.021 | A34 | 399.937 |
| A13 | −228.775 | A35 | 430.765 |
| A14 | −198.575 | A36 | 460.694 |
| A15 | −168.768 | A37 | 493.722 |
| A16 | −139.211 | A38 | 539.383 |
| A17 | −108.862 | A39 | 555.935 |
| A18 | −91.643 | A40 | 579.951 |
| A19 | −50.815 | A41 | 608.860 |
| A20 | −20.264 | A42 | 628.012 |
| A21 | 15.441 | A43 | 649.772 |
| A22 | 37.917 | | |

Table 3 shows that the distances along the x-axis are approximate for each two adjacent points, about 20–30 m. The approximate distances provide good conditions for understanding the subsidence process along the x-axis.

The observation points are surveyed using the global positioning satellites (GPS) method from 13 June 2018 to 15 July 2020, for 19 phases in total. The mean error of the GPS surveying method is deemed as 4 mm in this research [44]. Based on the x-coordinates and the GPS monitoring data of the observation points, the subsidence curves can be drawn to show the subsidence process along the x-axis, as shown in Figure 4.

Figure 4 shows the subsidence process of the observation points along the strike section of the workface. The subsidence values increase with the workface propulsion for all points. As for the subsidence curves, they go through a falling and rising process for all the phases, especially in the later phases. Of course, there are some outliers, whose values do not obey the common rules of change. For example, there are a few points whose subsidence values are apparently higher than 0 mm. Moreover, some curves present evident undulations along the strike section, such as the curves on 6 July 2018 and 24 October 2018.

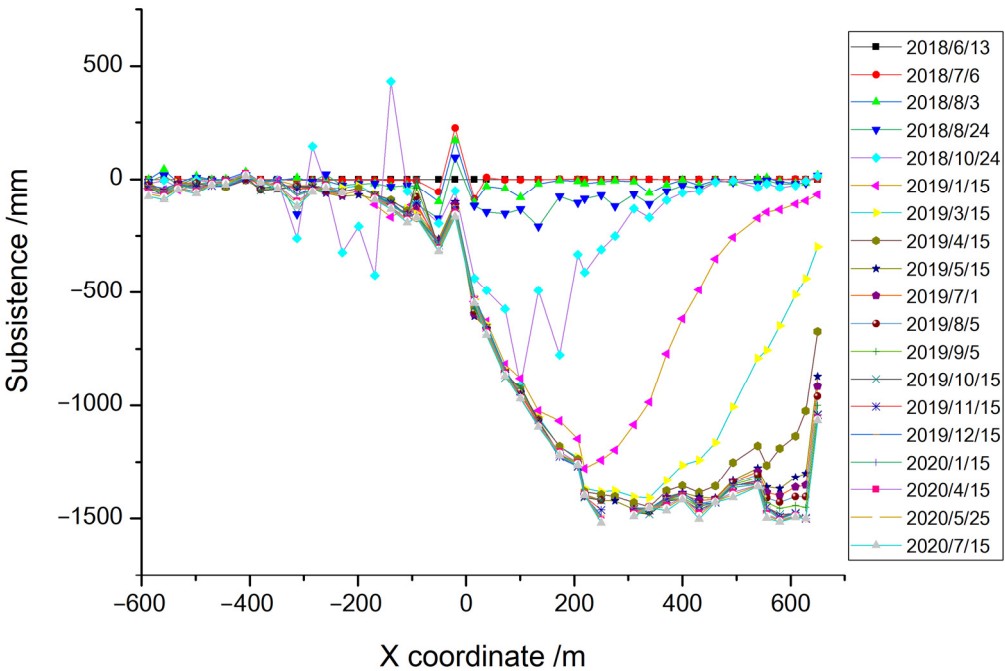

**Figure 4.** The subsidence curves along the strike section at each phase.

The influence radius (*r*) can be computed using the following equation [45]:

$$r = H/tan(\beta) \tag{1}$$

In Equation (1), *H* refers to the mining depth, and *β* refers to the influence angle.

Through computation, the influence radius is shown to be about 216.667 m in the workface. Meanwhile, the maximum subsidence values are predicted for the observation points using the probability integral method [46].

Based on the influence radius, the observed and predicted subsidence values and the x-coordinates of the observation points, the typical observation points were selected along the strike section of the workface. They are A21, A24, A28, A29 and A36; the observed and predicted maximum subsidence values are shown in Table 4.

**Table 4.** The observed and predicted maximum subsidence values for the typical points (mm).

| Typical Point | Observed Subsidence | Predicted Subsidence |
|:---:|:---:|:---:|
| A21 | −606.5 | −837 |
| A24 | −972.0 | −1260 |
| A28 | −1408.0 | −1419 |
| A29 | −1519.9 | −1390 |
| A36 | −1431.6 | −1505 |

Table 4 shows that the observed and predicted subsidence values are approximate for some points, such as A28; meanwhile, others have evident deviation, such as A21 and A24. The predicted maximum subsidence values increase as the x-coordinates increase, which is not suitable for the observed maximum subsidence values.

The reasons why the five typical points were selected are as follows. The x-coordinate of A21 is approximate to 0 m, which, in theory, refers to the open–off cut position. The x-coordinate of A24 is 100.574 m, about half of the influence radius, so it has the highest horizontal deformation and curvature values. The X coordinate of A28 is 219.384 m, which is approximate to the influence radius. A29 has the highest observed subsidence value among all the points. As for A36, it has the highest predicted subsidence value of all the points.

### 3.2. Mountainous Time Function Method

The mountainous time function method used in this research applies the time function method in conjunction with the mountainous conditions. It is the time function method applied to mountainous regions.

The time function method was presented by Knothe in 1953 [7,11]. Its equations are as follows:

$$W(t) = W_0 \varphi(t) \tag{2}$$

$$W_0 = mq\cos\alpha \tag{3}$$

$$\varphi(t) = 1 - e^{-c*t} \tag{4}$$

$$c = -\frac{v}{L_1} \ln 0.02 \tag{5}$$

In Equations (2)–(5), $W(t)$ is the subsidence value corresponding to the time $t$; $W_0$ is the final subsidence value; $\varphi(t)$ is the time function; $m$ is the mining height; $q$ is the subsidence coefficient; $\alpha$ is the dig angle of the coal seam; $c$ is the time influence parameter; $v$ is the advance rate, which can be calculated with Table 2; $L_1$ is the critical mining dimension, which is 100 m in this research.

The above time function method (Equations (2)–(5)) is suitable for flat topography. If it is used in a mountainous region, the subsidence value at coordinate $x$ can be transformed as the following equations [23]:

$$W'(x) = W(x)F(x) \tag{6}$$

$$F(x) = 1 + D_x \left(1 + A \cdot e^{-\frac{1}{2}(\frac{x}{r}+p)^2} + W_m \cdot e^{-t(\frac{x}{r}+p)^2}\right) \tan^2\alpha \tag{7}$$

Some parameters of Equations (6) and (7) are explained above, such as $r$. As for the other parameters, $W'(x)$ is the subsidence value at the observation point due to coal mining in a mountainous area; $W(x)$ is the subsidence value in flat ground with same geological conditions; $F(x)$ is used to simplify the Equation (6); $D_x$ is the ground characteristic coefficient, which is determined by the characteristics at the location of the observation points, such as the slope, curvature, land use, geological conditions and so on; $W_m$ is the maximum subsidence value; $\alpha$ is the inclination angle of the ground trend surface. $A$, $P$ and $t$ are slip influence parameters, whose values are determined to be $2\pi$, 2 and $\pi$, respectively, in this research.

As shown in Figure 3, there are 19 observation phases in this research. Therefore, the working face can be divided into 18 elements. The propulsion status, including the propulsion distance and the advance rate, are computed in Table 5.

**Table 5.** The propulsion status in different phases.

| Phase | Propulsion Distance (m) | Advance Rate (m/d) | Phase | Propulsion Distance (m) | Advance Rate (m/d) |
|---|---|---|---|---|---|
| 13 June 2018 | 0 | 0 | 5 August 2019 | 87.23 | 2.49 |
| 6 July 2018 | 13.47 | 0.59 | 5 September 2019 | 96.13 | 3.10 |
| 3 August 2018 | 64.72 | 2.31 | 15 October 2019 | 104.15 | 2.60 |
| 24 August 2018 | 60.15 | 2.86 | 15 November 2019 | 86.12 | 2.78 |
| 24 October 2018 | 159.59 | 2.62 | 15 December 2019 | 80.39 | 2.68 |
| 15 January 2019 | 195.07 | 2.35 | 15 January 2020 | 70.17 | 2.26 |
| 15 March 2019 | 167.72 | 2.84 | 15 April 2020 | 237.85 | 2.61 |
| 15 April 2019 | 94.08 | 3.03 | 25 May 2020 | 64.35 | 1.61 |
| 15 May 2019 | 95.78 | 3.19 | 15 July 2020 | 8.05 | 0.16 |
| 1 July 2019 | 134.19 | 2.86 | | | |

In Table 5, the propulsion distance is the mining length between the current phase and the last phase, and the advance rate is the propulsion distance divided by the days during the two phases. From Table 5, we can see that the propulsion distance varies significantly at different phases. For example, it is 237.85 m on 15 April 2020, while 13.47 m on 6 July 2018. Meanwhile, the advance rate is also different at different phases.

For the 18 elements of the workface, the maximum progressive subsidence can be computed for the moment $t$ using the following equation [7,23]:

$$W\prime_n(t - t_1 - t_2 - \ldots - t_{n-1}) = W_0 F(x)\varphi(t - t_1 - t_2 \ldots - t_{n-1}) \tag{8}$$

Using the superposition principle, the progressive subsidence of the observation points along the strike section with $x$ coordinate at time $t$ can be computed using the following equations:

$$
\begin{aligned}
W(x, t) = (1 - e^{-ct}) \quad & (W(x)F(x) - W(x - v_1 t_1)F(x - v_1 t_1)) \\
& + \left(1 - e^{-c(t-t_1)}\right)(W(x - v_1 t_1)F(x - v_1 t_1) - W(x - v_1 t_1 - v_2 t_2)F(x - v_1 t_1 - v_2 t_2)) \\
& + \left(1 - e^{-c(t-t_1-t_2)}\right)(W(x - v_1 t_1 - v_2 t_2)F(x - v_1 t_1 - v_2 t_2) \\
& - W(x - v_1 t_1 - v_2 t_2 - v_3 t_3)F(x - v_1 t_1 - v_2 t_2 - v_3 t_3)) + \ldots \\
& + \left(1 - e^{-c(t-t_1-t_2\ldots-t_{n-1})}\right)(W(x - v_1 t_1 - v_2 t_2 - \ldots - v_{n-1} t_{n-1})F(x - v_1 t_1 - v_2 t_2 - \ldots - v_{n-1} t_{n-1}) \\
& - W(x - v_1 t_1 - v_2 t_2 - \ldots - v_n t_n)F(x - v_1 t_1 - v_2 t_2 - \ldots - v_n t_n))
\end{aligned} \tag{9}
$$

where

$$W(x) = \frac{W_m}{2}\left(1 + \frac{2}{\sqrt{\pi}}\int_0^{\frac{\sqrt{\pi}}{r}x} e^{-u^2} du\right) \tag{10}$$

Based on Equations (9) and (10), the subsidence process can be predicted at any time by using the parameters in the workface. This is the mountainous time function method.

### 3.3. EnKF Method

The mountainous time function method can predict the subsidence process of the observation points along the strike section at any time. Meanwhile, the subsidence values at the locations of the observation points are measured using the GPS method in 19 phases. To improve the prediction accuracy, it is critical to assimilate the GPS observation data into the prediction process.

As a sequential assimilation method, the Kalman filter method is widely used in acquiring the least square estimation of a system state [47]. The covariance matrix of this method is updated using a linear model, so its use is limited to linear models with small scales. To solve this problem, the EnKF method is presented based on the random forecast theory [28,38]. The EnKF method estimates the statistical moment of the system state through constructing samples, so as to assimilate the data with different sources. Hence, the EnkF method is widely used in many fields with large scale and non-linear problems, such as weather forecasting, ocean dynamics, hydrology, and so on [48,49].

As a method of estimating analysis error covariance and background error covariance using the ensemble, the basic idea of the EnKF method is as follows [50]: through initializing a set of system state samples as the background ensemble, the analysis ensemble can be achieved by updating each element of the background ensemble using observation information, so as to estimate the real mean and covariance values of the state. Then, the background ensemble at the next moment can be acquired by transferring the sample ensemble through the system model. The specific computation process of the EnKF method in this research is as follows:

Firstly, the state and measuring equations are listed as follows:

$$X_k = AX_{k-1} + BU_k + W_k \tag{11}$$

$$Z_k = HX_k + V_k \tag{12}$$

In Equations (11) and (12), $X_k$ and $X_{k-1}$ are the state variables at times $k$ and $k-1$, respectively; $A$ and $B$ are the parameter matrix of the system state; $U_k$ is the controlling value of the system; $Z_k$ is the measuring vector; $H$ is the parameter matrix of the measuring equation, which is a unit matrix in this research; $W_k$ and $V_k$ are the system error vector and

measuring error vector, whose covariances are determined as $Q$ (25 mm$^2$) and $R$ (16 mm$^2$), respectively, in this research.

The initial state ensemble can be acquired by adding stochastic disturbance to the initial state variables:

$$W(x_i, t_0) = W(x, t_0) + u_i, u_i \sim N(0, R) \tag{13}$$

In Equation (13), $u_i$ is the background error, which is subject to the normal distribution with a mean of 0 and a variance of $R$.

Then, the state variable prediction is conducted based on the mountainous time function method, which is derived from Equation (9) and shown as the following:

$$
\begin{aligned}
W(x, t_k) = W(x, t_{k-1})' \quad &+ \left(1 - e^{-c(t_k - t_{k-1})}\right) \left(e^{-ct_{k-1}}(W(x)F(x) - W(x - v_1 t_1)F(x - v_1 t_1)) \right. \\
&+ e^{-c(t_{k-1} - t_1)}(W(x - v_1 t_1)F(x - v_1 t_1) - W(x - v_1 t_1 - v_2 t_2)F(x - v_1 t_1 - v_2 t_2)) \\
&+ e^{-c(t_{k-1} - t_1 - t_2)}(W(x - v_1 t_1 - v_2 t_2)F(x - v_1 t_1 - v_2 t_2) \\
&- W(x - v_1 t_1 - v_2 t_2 - v_3 t_3)F(x - v_1 t_1 - v_2 t_2 - v_3 t_3)) \\
&+ \ldots e^{-c(t_{k-1} - t_1 - t_2 \ldots - t_{k-2})}(W(x - v_1 t_1 - v_2 t_2 - \ldots - v_{k-2} t_{k-2})F(x - v_1 t_1 - v_2 t_2 - \ldots - v_{k-2} t_{k-2}) \\
&\left. - W(x - v_1 t_1 - v_2 t_2 - \ldots - v_{k-1} t_{k-1})F(x - v_1 t_1 - v_2 t_2 - \ldots - v_{k-1} t_{k-1})\right) + \omega_i, \omega_i \sim N(0, Q)
\end{aligned}
\tag{14}
$$

The state variable can be predicted using Equation (14). In this equation, $W(x, t_k)$ is the predicted ensemble at the moment $t_k$ acquired from the analyzed subsidence value at moment $t_{k-1}$; $W(x, t_{k-1})'$ is the analyzed ensemble at the moment $t_{k-1}$, computed from the observation information at moment $t_{k-1}$; $\omega_i$ is the model error, which is subject to the normal distribution with a mean of 0 and a variance of $Q$.

The ensemble error covariance ($P_e$) is computed using the following equation:

$$P_e = \frac{A'A'^T}{N - 1} \tag{15}$$

In Equation (15), $T$ is the transpose symbol; $A' = A - \overline{A}$ is the ensemble disturbance; $\overline{A}$ is the mean value of the ensemble; $N$ is the number of ensemble elements.

The gain matrix $K$ can be acquired using the following equation:

$$K = P_e * H^T * \left(H * P_e * H^T + R\right)^{-1} \tag{16}$$

The state variable can be updated using the following equation:

$$W(x, t_k)' = W(x, t_k) + K(D_k - H * W(x, t_k)) \tag{17}$$

In Equation (17), $W(x, t_k)'$ is the analyzed ensemble at moment $t_k$ computed from the observation information at moment $t_k$; $D_k$ is the observation information at moment $t_k$.

Finally, the ensemble error covariance is updated using the following equation:

$$P_a = P_e - H^T \left(HP_e H^T + R_e\right)^{-1} HP_e \tag{18}$$

In Equation (18), $P_a$ is the updated ensemble error covariance; $P_e$ is the predicted ensemble error covariance; $R_e$ is the observed error covariance.

Overall, the larger the number of the set is, the better the assimilation quality is. Referencing the relevant literature, the number of the set was determined as 100 in this research [48–50].

Through the iteration computation process from Equations (13)–(18), the ground subsidence of the observation points can be predicted using the EnKF assimilation method. It is executed using Matlab R2014b software in this research.

*3.4. Error Indexes Selection*

Based on the relevant research [10,26,51], four error indexes were selected to quantify the prediction quality. They are the mean error (*ME*), the mean absolute error (*MAE*), the root mean square error (*RMSE*) and the mean absolute percentage error (*MAPE*). The equations for these error indexes are as follows:

$$ME = \frac{\sum_{i=1}^{i=n}(x_i - y_i)}{n} \tag{19}$$

$$MAE = \frac{\sum_{i=1}^{i=n}|x_i - y_i|}{n} \tag{20}$$

$$RMSE = \sqrt{\frac{\sum_{i=1}^{i=n}\left[(x_i - y_i)^2\right]}{n}} \tag{21}$$

$$MAPE = \frac{1}{n}\sum_{i=1}^{i=n}\frac{|x_i - y_i|}{|y_i|} \times 100\% \tag{22}$$

In Equations (19)–(22), *n* is the total number of the values, which is 19 in this research, corresponding to 19 phases of the typical observation points; *x* is the prediction value acquired by both the mountainous time function and the EnKF methods; *y* is the real value, which is regarded as the assimilated value acquired by the EnKF method.

## 4. Results

Based on the mountainous time function and the EnKF method, the ground subsidence values of the typical points were modeled and predicted for 19 phases. To evaluate the prediction results, the error curves and error indexes are acquired accordingly.

*4.1. Prediction Results Using the Mountainous Time Function and EnKF Methods*

The ground subsidence prediction and assimilation are conducted for five typical observation points along the strike section: A21, A24, A28, A29 and A36.

4.1.1. Prediction Results of Observation Point A21

Observation point A21 is approximate to the open–off cut position. The ground subsidence values of A21 were measured at 19 phases using the GPS method. Meanwhile, the ground subsidence values of A21 were predicted using both the mountainous time function and the EnKF method. Furthermore, the ground subsidence values were assimilated by incorporating the measured observation information into the model simulation data using the EnKF method.

Hence, the subsidence values of observation point A21 have four data sources: GPS measurement, mountainous time function prediction, EnKF prediction and assimilation. The GPS data was acquired by field survey, whereas the other three data sources were generated from the field data and the methods.

The results of the subsidence values of A21 acquired by the four methods are shown in Table A1 in the Appendix A.

Based on Table A1, the subsidence curves of A21 were drawn using different methods, which are shown in Figure 5.

Table A1 and Figure 5 show the mountainous time function predicted values, and the EnKF predicted and assimilated values are approximate to the measured values on the whole. Specifically, there are evident deviations at some phases, such as 3 August 2018, 24 August 2018 and 15 May 2019. On 3 August 2018, the measured value is −103.2 mm, whereas the mountainous time function predicted values, the EnKF predicted values and the assimilated values are −167.5 mm, −225.8 mm and −210.1 mm, respectively. These large deviations result from the abrupt change during this phase. The measured values are 0.0 mm on 13 June 2018, and −86.6 mm on 2018/7/6, which changes to −103.2 mm by

3 August 2018. As such, the predicted and assimilated values have difficulty in adapting immediately to this abrupt change.

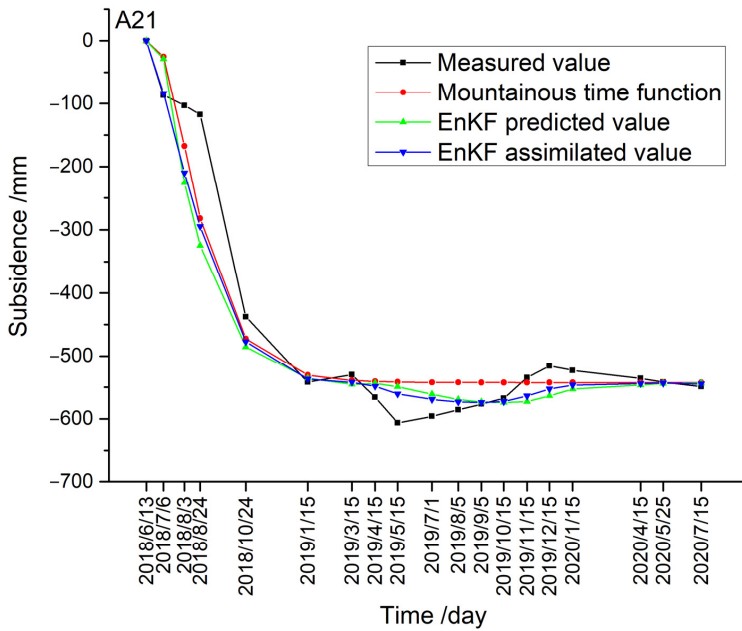

**Figure 5.** Subsidence curves of A21 using different methods.

### 4.1.2. Prediction Results of Observation Point A24

Observation point A24 is at half of the influence radius, which has the highest horizontal deformation and curvature values. Similar to those of A21, the ground subsidence values of A24 can be acquired from four sources: GPS measurement, mountainous time function prediction, EnKF prediction and assimilation. These values were acquired from 19 phases from 13 June 2018 to 15 July 2020.

The results of the subsidence values of A24 acquired in 19 phases and by the four methods are shown in Table A2 of Appendix A.

Based on Table A2, the subsidence curves of A24 were drawn using different methods, which are shown in Figure 6.

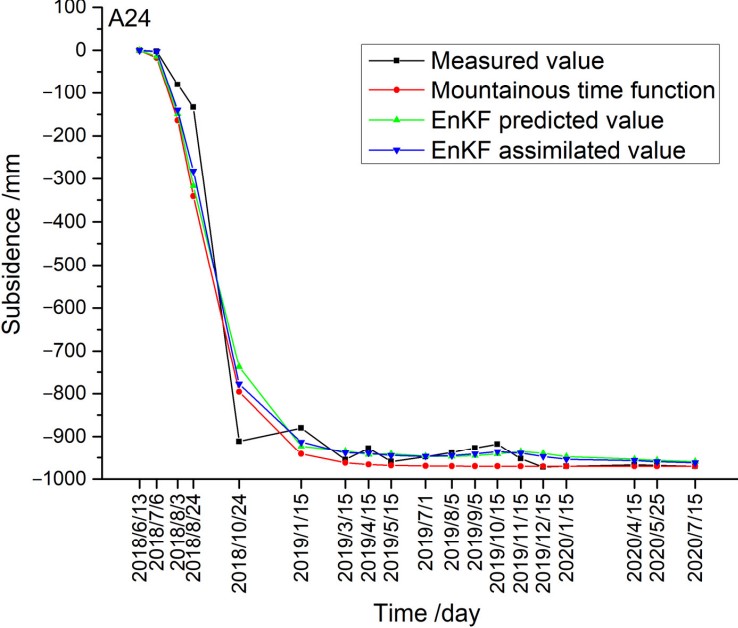

**Figure 6.** Subsidence curves of A24 using different methods.

From Table A2 and Figure 6, we can see that, similarly to A21, the mountainous time function's predicted values and the EnKF-predicted and -assimilated values of A24 are approximate to the measured values on the whole. Specifically, there are evident deviations at some phases, such as on 3 August 2018, 24 August 2018 and 24 October 2018. On 24 August 2018, the measured value is −133.5 mm, whereas the mountainous time function-predicted values, and the EnKF-predicted and -assimilated values are −340.8 mm, −317.1 mm and −282.4 mm, respectively. These large deviations result from the abrupt change during these phases, and the predicted and assimilated values have difficulty in adapting immediately to this abrupt change.

### 4.1.3. Prediction Results of Observation Point A28

Observation point A28 is approximate to the influence radius. Similar to those of A24, the subsidence values of observation point A28 also have four data sources: GPS measurement, mountainous time function prediction, EnKF prediction and assimilation. These values were acquired over 19 phases, from 13 June 2018 to 15 July 2020. The results of the subsidence values of A28 acquired over 19 phases and by the four methods are shown in Table A3 of Appendix A.

Based on Table A3, the subsidence curves of A28 were drawn using different methods, which are shown in Figure 7.

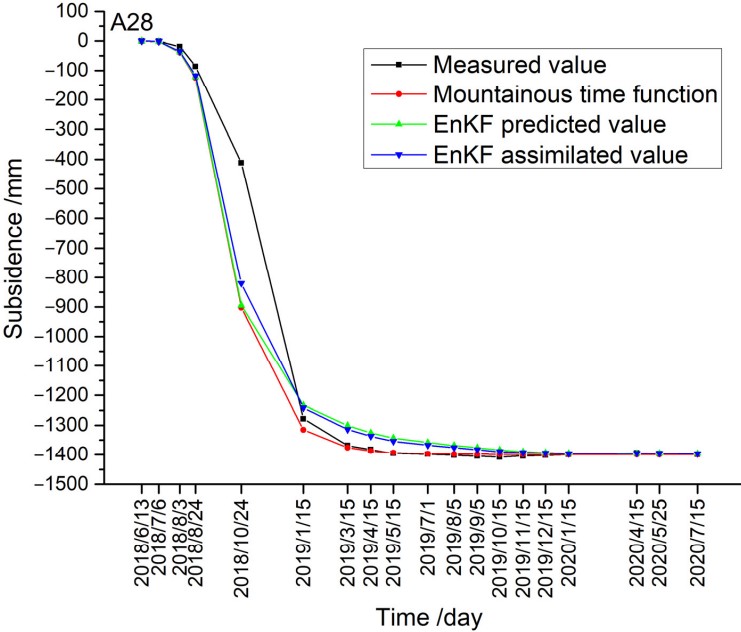

**Figure 7.** Subsidence curves of A28 using different methods.

Table A3 and Figure 7 show that the mountainous time function-predicted values and the EnKF-predicted and -assimilated values are approximately equal to the measured values, on the whole. Specifically, there is evident deviation on 24 October 2018. On 24 October 2018, the measured value is −411.9 mm, whereas the mountainous time function predicted values and the EnKF predicted and assimilated values are −902.5 mm, −894.7 mm and −820.2 mm, respectively. The large deviation may be due to the fact that the measured value does not adapt to the theoretical values acquired by the mountainous time function and the EnKF method.

### 4.1.4. Prediction Results of Observation Point A29

Observation point A29 has the highest observed subsidence value among all the points. The subsidence values of A29 were measured, predicted and assimilated across 19 phases. The results of the subsidence values of A29 are shown in Table A4 of Appendix A.

Based on Table A4, the subsidence curves of A29 were drawn using different methods, which are shown in Figure 8.

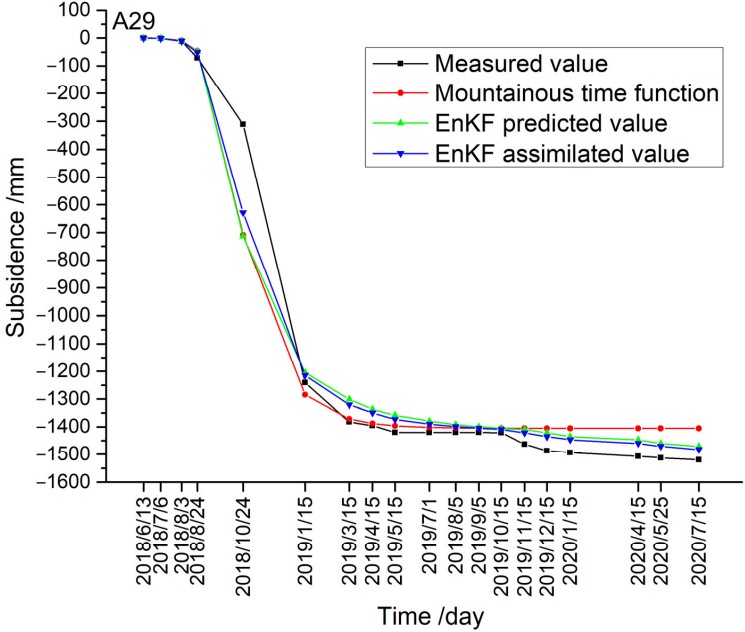

**Figure 8.** Subsidence curves of A29 using different methods.

Table A4 and Figure 8 show that the mountainous time function-predicted values and the EnKF-predicted and -assimilated values are approximate to the measured values on the whole. Specifically, there is evident deviation on 24 October 2018. On 24 October 2018, the measured value is −311.1 mm, whereas the mountainous time function-predicted values and the EnKF-predicted and -assimilated values are −708.5 mm, −713.6 mm and −629.0 mm, respectively. Meanwhile, there are evident deviations at the end of the subsidence process. Taking the last phase 15 July 2020 as an example, the measured value is −1519.9 mm, whereas the mountainous time function predicted values, the EnKF predicted and assimilated values are −1405.3 mm, −1470.8 mm and −1482.4 mm, respectively.

4.1.5. Prediction Results of Observation Point A36

Observation point A36 has the highest predicted subsidence value of all the points. This prediction value was acquired using the integral probability method, which is the final subsidence value. The subsidence process was measured using the GPS surveying method, predicted by the mountainous time function and EnKF method, and assimilated by the EnKF method in 19 phases from 13 June 2018 to 15 July 2020.

Accordingly, the values of the subsidence process of observation point A36 acquired in the 19 phases and by the four methods are shown in Table A5 in the Appendix A section.

Based on Table A5, the subsidence curves of A36 were drawn using different methods, as shown in Figure 9.

From Table A5 and Figure 9 we can see that the mountainous time function-predicted values and the EnKF-predicted and -assimilated values are approximate to the measured values on the whole. Specifically, there is evident deviation on 15 January 2019. On 15 January 2019, the measured value is −352.0 mm, whereas the mountainous time function predicted values and the EnKF-predicted and -assimilated values are −830.1 mm, −825.1 mm and −731.7 mm, respectively. These large deviations area result of the abrupt change in this phase, as the predicted and assimilated values had difficulty in adapting immediately to this change.

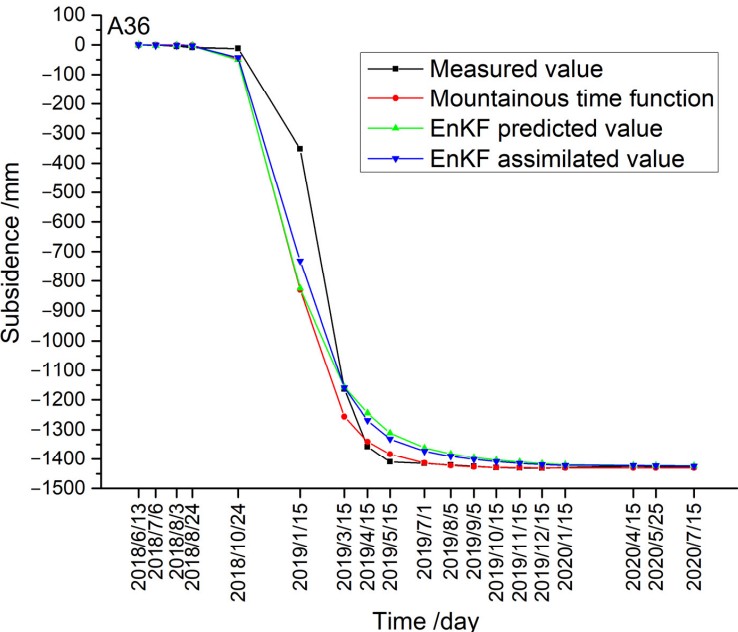

**Figure 9.** Subsidence curves of A36 using different methods.

## 4.2. Error Curves Using the Mountainous Time Function and EnKF Methods

The prediction results show that they are approximate to the measured values on the whole, but they have evident deviations at some phases for the typical points. As the measured values contain errors (with error covariance of 16 mm$^2$), the assimilated values are taken as the real values in this research. The assimilation results also contain errors, although they are the most accurate data. Taking the assimilation results as the real values, the error values can be computed using both the mountainous time function and EnKF methods, which are presented as the curves for the typical observation points and are shown in the following section.

### 4.2.1. Error Curves Using the Mountainous Time Function Method

Taking the assimilated results of the EnKF method as the real values, the error values of the prediction results acquired using the mountainous time function method were computed for the five typical observation points. The curves of these error values are presented in Figure 10.

Figure 10 shows the error value ranges from about −80 mm on 15 July 2020 for A29 to about 100 mm on 15 January 2019 for A36. Excluding A29, the error values of the other four observation points ultimately approach 0 mm. Hence, the absolute error values of the four observation points undergo an increase first, and then show a decreasing tendency.

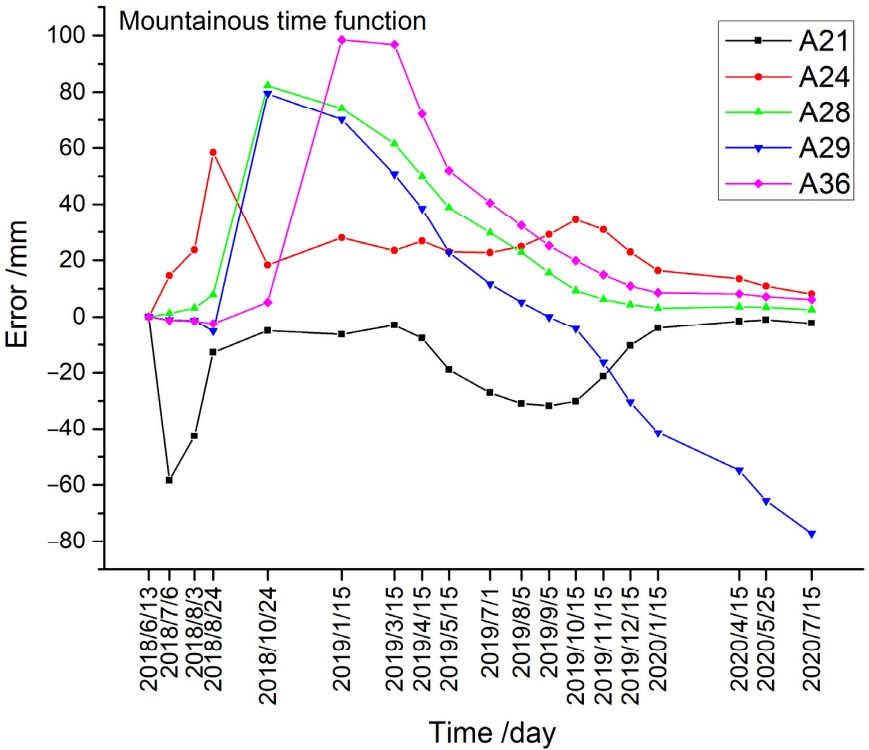

**Figure 10.** Error curves of the typical points using the mountainous time function method.

### 4.2.2. Error Curves Using the EnKF Method

Taking the assimilated results as real values, the error values of the prediction results acquired using the EnKF method were also computed for the five typical observation points. The curves of these error values are presented in Figure 11:

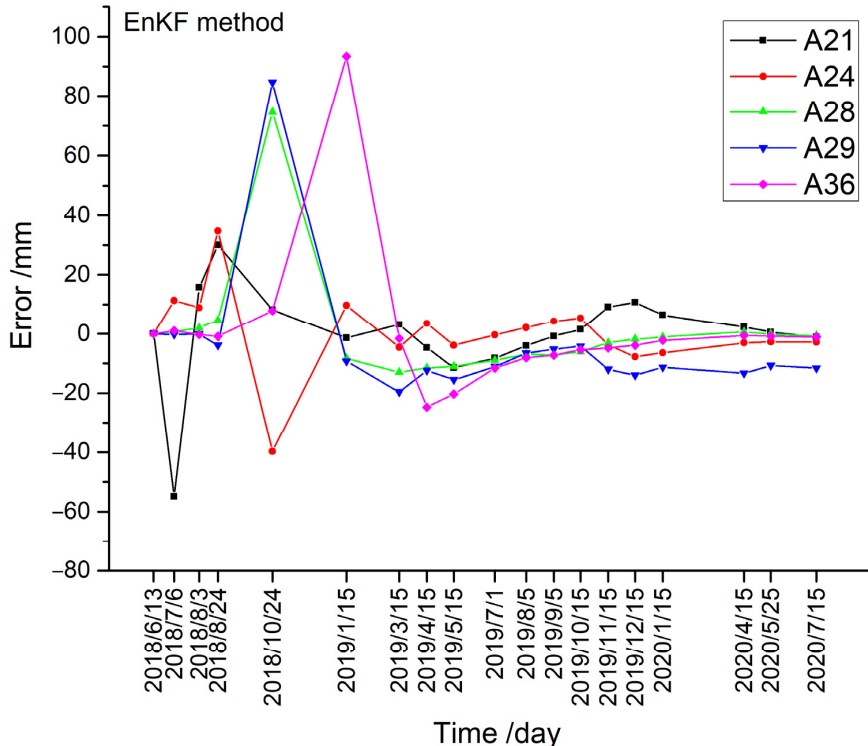

**Figure 11.** Error curves of the typical points using the EnKF method.

*4.3. Error Indexes Using the Mountainous Time Function and EnKF Methods*

To quantify the error values, the selected four error indexes were computed for the mountainous time function method and EnkF method. Meanwhile, the average values of the error indexes were computed and evaluated.

4.3.1. Computation of Error Indexes Using the Mountainous Time Function Method

Based on Tables 5 and A1–A4, the four error indexes were computed for the typical five observation points predicted using the mountainous time function method. The computation results are presented in Figure 12.

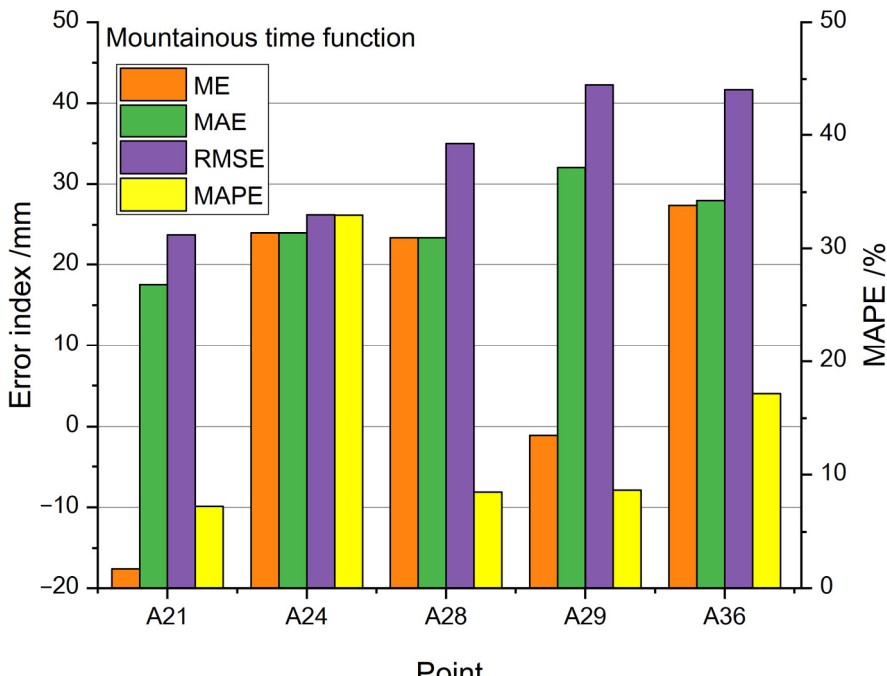

**Figure 12.** Error indexes of the typical points using the mountainous time function method.

Figure 12 shows that the values of the error indexes are different for the five typical observation points. For example, the *ME*, *MAE*, *RMSE* and *MAPE* values for A21 are −17.5 mm, 17.5 mm, 23.7 mm and 7.2%, respectively, which are apparently lower than those for the other four points. Observation point A36 has the highest *ME* and *MAPE* values, and relatively high values for the other two error indexes, which are 27.4 mm, 28.0 mm, 41.7 mm and 17.2%, respectively. Meanwhile, observation point A29 has the highest *RMSE* and *MAE* values, which are 42.3 mm and 32.0 mm, respectively.

4.3.2. Computation of Error Indexes Using the EnKF Method

Based on Tables 5 and A1–A4, the four error indexes were computed for the five typical observation points predicted using the EnKF method. The computation results are presented in Figure 13.

From Figure 13, we can see that the error index values using the EnKF method appear to be lower than those acquired using mountainous the time function method. A29 has the lowest *ME* value (−4.2 mm) and *MAPE* value (2.9%), whereas the *ME* values for the other four points are approximate to 0 mm. As for the *MAE* values, they are about 10 mm for all the points. A36 has the highest *RMSE* value (23.7 mm), and A24 has the highest *MAPE* value (24.0%).

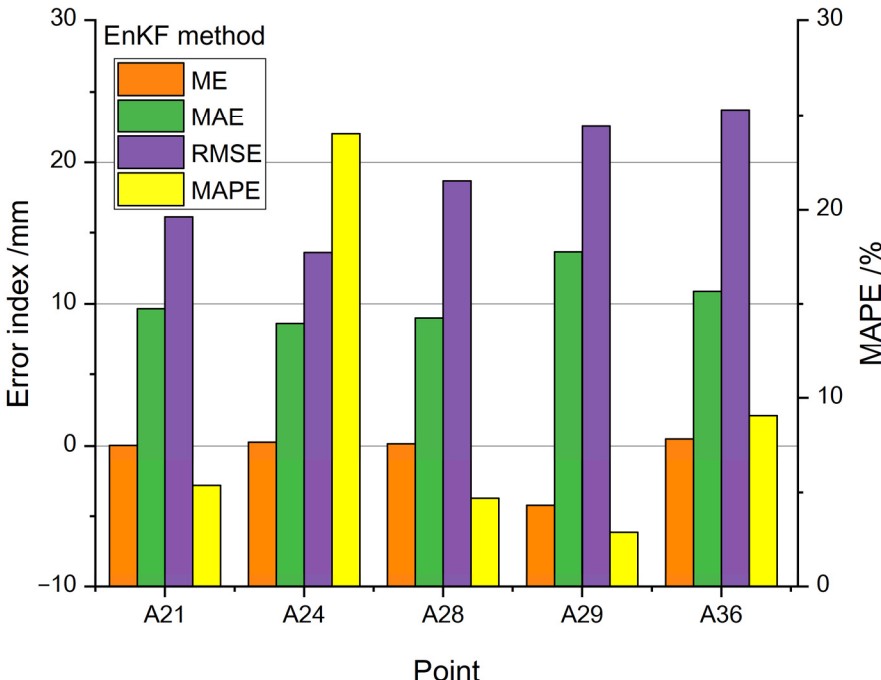

**Figure 13.** Error indexes of the typical points using EnKF method.

4.3.3. Average Values of the Error Indexes

To quantify the values of the error indexes, the average values were computed for the mountainous time function and the EnKF method. The computation results are shown in Table 6.

**Table 6.** Average values of the error indexes for the two methods.

| Method | *ME*/mm | *MAE*/mm | *RMSE*/mm | *MAPE*/% |
|---|---|---|---|---|
| Mountainous time function method | 11.2 | 24.9 | 33.8 | 14.9 |
| EnKF method | −0.7 | 10.4 | 19.0 | 9.2 |

Table 6 shows that the *ME*, *MAE*, *RMSE* and *MAPE* values are 11.2 mm, 24.9 mm, 33.8 mm and 14.9%, respectively, for the mountainous time function method, whereas they are −0.7 mm, 10.4 mm, 19.0 mm and 9.2 mm, respectively, for the EnKF method. Hence, the prediction accuracy is largely improved by using the EnKF method. This is because the observation data are assimilated into the prediction process. As such, the presented method used in this research is meaningful and valuable.

**5. Discussion**

*5.1. Innovations*

This research presents a novel ground subsidence prediction method, the EnKF method, by taking the mountainous time function method as the state equation. Taking a workface of the TunLan Minefield of the Xishan Coalfield in the Shanxi Province of China as the study area, the ground subsidence processes were predicted for five typical observation points using both the EnKF and mountainous time function methods. Finally, the prediction performance was evaluated using error curves and error indexes for the two methods. The evaluation results show that the prediction accuracy of the EnKF method is much higher than that of the mountainous time function. The high accuracy of the EnKF method can be attributed to the fact that it assimilates the observation information in the prediction process. Based on this research, the following innovations were achieved:

A novel ground subsidence prediction method was presented by incorporating the mountainous time function method into the EnKF method as the state equation method. Previous research have mostly assumed the ground to have a flat topography when using time function methods [7,9,26]. In this research, the mountainous time function method was used to predict the ground subsidence process in mountainous regions by using the ground characteristic coefficient, which is determined by slope, curvature, land use and geological factors [21–23]. Hence, the mountainous time function performs better than the normal time function method in ground subsidence predictions. Furthermore, the iteration equation was derived from the mountainous time function method, which became the state equation of the EnKF method [28,32,34]. The derived iteration equation was first acquired in this research. It can be used in various relevant research and methodological applications [29–31].

The performance of the prediction methods used in the ground subsidence process have been evaluated and compared. Zhang and Zhang acquired *ME*, *RMSE* and *MAPE* values of 91.9 mm, 133.4 mm and 12.2% for the time function method, and 3.1 mm, 78.3 mm and 7.2% for the Kalman filter method [9]. Cui et al. judged that the *MAPE* value was about 8% by using the time function method [7]. Zhang et al. computed an *MAPE* value of 23% for the classical time function method [3]. Cheng et al. acquired the *MAPE* value of 23.1 for the Knothe time function method [15]. The *RMSE* value of the total Kalman filter method is about 100 mm [26]. In river network data assimilation, the maximum prediction error for the flood stage was no more than 0.02 m when using the EnKF method [50]. As ground subsidence predictions are conducted in mountainous regions in this research, the high performance of the prediction methods has both theoretical and practical significance.

### 5.2. Prospects

This research presents a novel method for ground subsidence prediction. In addition to subsidence, there are other ground deformation factors, such as inclination, horizontal movement, horizontal deformation, curvature, and so on. Additionally, this research takes a workface of the Tunlan Minefield as the experimental area. Future research can be conducted at other relevant workfaces or in other similar regions for verification.

In addition to the EnKF method and the mountainous time function method, there are many other methods for predicting the ground subsidence process in mining areas. As for the time function method, there is the Knothe time function [11], the inverse function [15], the logistic function [16], the arc tangent function [17], the negative exponential function, the Weibull time sequence function [18], the Bertalanffy time function [19], the normal time function [20], the structural backfill method [52] and so on. As well as the EnKF method, there are many other Kalman filter methods, such as the classical Kalman filter [7,9], the extended Kalman filter [52,53], the unscented Kalman filter [54,55], and the total Kalman filter [56,57]. Furthermore, the InSAR and GPS surveying methods play important role in the subsidence monitoring and prediction [58–61]. These methods can also be applied to and explored in ground subsidence prediction in future research.

The presented method is applied in ground subsidence due to the focus of this research being coal mining. In other regions, there are many other factors that result in ground subsidence, such as large groundwater exploitation in urban areas [62–64], the permafrost free-thawing process in the Qinghai-Tibet Plateau, and so on [65–67].

### 6. Conclusions

According to this research, the following conclusions can be reached:

(1) The EnKF method was constructed by taking the mountainous time function method as the state equation. Based on both the mountainous time function and the EnKF methods, ground subsidence processes were predicted in a workface of the Tunlan Minefield of the Xishan Coalfield of China. Furthermore, the prediction results were assimilated using the EnKF method. Taking the assimilation results as the real values, the errors of the prediction results were evaluated and compared for the two methods.

(2)　The prediction results of the two methods and the assimilation results acquired by the EnKF method are, on the whole, approximately equal to the measured values obtained by GPS monitoring. At some phases, there are evident deviations for the typical observation points. Due to undulating topography and complex topographic ground conditions, there are abrupt changes during these phases. These large deviations result from these abrupt changes, and the predicted and assimilated values have difficulty in adapting to these changes immediately. In addition, the deviations differ largely between the various observation points.

(3)　Taking the assimilation results acquired by the EnKF method as the real values, the error values range from about −80 mm to about 100 mm for the mountainous time function method, whereas they range from about −60 mm to about 100 mm for the EnKF method. Except for point A29 in the mountainous time function method, all of the error values converge to 0 mm. The average values for the *ME*, *MAE*, *RMSE* and *RMSE* indexes are 11.2 mm, 24.9 mm, 33.8 mm and 14.9%, respectively, for the mountainous time function method, whereas they are −0.7 mm, 10.4 mm, 19.0 mm and 9.2%, respectively, for the EnKF method. This large improvement arises from the fact that the observation data were assimilated into the prediction process for the EnKF method.

**Author Contributions:** Conceptualization, J.Z.; Methodology, S.Z.; Software, S.Z.; Validation, S.Z. and J.Z.; Formal analysis, S.Z. and J.Z.; Investigation, S.Z.; Resources, J.Z.; Data curation, S.Z.; Writing—original draft preparation, S.Z.; Writing—review and editing, J.Z.; Visualization, S.Z.; Supervision, J.Z.; Project administration, J.Z.; Funding acquisition, J.Z. All authors have read and agreed to the published version of the manuscript.

**Funding:** This work was funded by the National Natural Science Foundation of China [Grant Number: 42171424, 42271432].

**Data Availability Statement:** The data used to support the findings of this study are available from the corresponding author upon request.

**Conflicts of Interest:** The authors declare no conflict of interest.

## Appendix A

**Table A1.** The subsidence values of the observation point A21 acquired by different methods (mm).

| Phase | Measured Value | Mountainous Time Function Value | EnkF Predicted Value | EnkF Assimilated Value |
|---|---|---|---|---|
| 13 June 2018 | 0.0 | 0.0 | 0.0 | 0.0 |
| 6 July 2018 | −86.6 | −25.2 | −28.7 | −83.6 |
| 3 August 2018 | −103.2 | −167.5 | −225.8 | −210.1 |
| 24 August 2018 | −117.4 | −282.0 | −324.7 | −294.8 |
| 24 October 2018 | −437.9 | −472.3 | −485.4 | −477.3 |
| 15 January 2019 | −542.0 | −530.8 | −535.8 | −537.2 |
| 15 March 2019 | −530.2 | −539.3 | −545.4 | −542.3 |
| 15 April 2019 | −566.0 | −540.9 | −543.9 | −548.6 |
| 15 May 2019 | −606.5 | −541.7 | −549.2 | −560.6 |
| 1 July 2019 | −596.0 | −542.3 | −561.1 | −569.3 |
| 5 August 2019 | −585.9 | −542.4 | −569.4 | −573.5 |
| 5 September 2019 | −576.6 | −542.5 | −573.5 | −574.3 |
| 15 October 2019 | −567.6 | −542.6 | −574.2 | −572.7 |
| 15 November 2019 | −534.8 | −542.6 | −572.9 | −563.8 |
| 15 December 2019 | −516.3 | −542.6 | −563.7 | −552.9 |
| 15 January 2020 | −523.2 | −542.6 | −553.0 | −546.6 |
| 15 April 2020 | −536.1 | −542.6 | −546.5 | −544.2 |
| 25 May 2020 | −541.8 | −542.6 | −544.2 | −543.6 |
| 15 July 2020 | −549.0 | −542.6 | −543.7 | −544.9 |

**Table A2.** The subsidence values of the observation point A24 acquired by different methods (mm).

| Phase | Measured Value | Mountainous Time Function Value | EnkF Predicted Value | EnkF Assimilated Value |
|---|---|---|---|---|
| 13 June 2018 | 0.0 | 0.0 | 0.0 | 0.0 |
| 6 July 2018 | −2.2 | −17.4 | −14.1 | −2.8 |
| 3 August 2018 | −80.4 | −164.0 | −149.2 | −140.3 |
| 24 August 2018 | −133.5 | −340.8 | −317.1 | −282.4 |
| 24 October 2018 | −911.5 | −795.9 | −737.8 | −777.6 |
| 15 January 2019 | −879.9 | −941.0 | −922.7 | −913.0 |
| 15 March 2019 | −954.8 | −962.0 | −934.0 | −938.5 |
| 15 April 2019 | −927.7 | −966.0 | −942.7 | −939.2 |
| 15 May 2019 | −959.2 | −968.0 | −941.1 | −945.0 |
| 1 July 2019 | −948.0 | −969.4 | −946.3 | −946.7 |
| 5 August 2019 | −937.2 | −969.8 | −947.0 | −944.9 |
| 5 September 2019 | −927.2 | −970.0 | −945.1 | −940.8 |
| 15 October 2019 | −917.6 | −970.1 | −941.1 | −935.7 |
| 15 November 2019 | −952.4 | −970.1 | −935.5 | −939.3 |
| 15 December 2019 | −972.0 | −970.2 | −939.6 | −947.3 |
| 15 January 2020 | −970.2 | −970.2 | −947.4 | −953.8 |
| 15 April 2020 | −966.9 | −970.2 | −953.6 | −956.7 |
| 25 May 2020 | −968.4 | −970.2 | −956.6 | −959.3 |
| 15 July 2020 | −970.2 | −970.2 | −959.2 | −962.0 |

**Table A3.** The subsidence values of the observation point A28 acquired by different methods (mm).

| Phase | Measured Value | Mountainous Time Function Value | EnkF Predicted Value | EnkF Assimilated Value |
|---|---|---|---|---|
| 13 June 2018 | 0.0 | 0.0 | 0.0 | 0.0 |
| 6 July 2018 | −1.2 | −2.5 | −2.0 | −1.2 |
| 3 August 2018 | −18.9 | −38.5 | −37.3 | −35.3 |
| 24 August 2018 | −86.3 | −125.4 | −122.1 | −117.5 |
| 24 October 2018 | −411.9 | −902.5 | −894.7 | −820.2 |
| 15 January 2019 | −1278.5 | −1315.5 | −1233.0 | −1241.3 |
| 15 March 2019 | −1368.8 | −1375.6 | −1301.1 | −1314.1 |
| 15 April 2019 | −1381.8 | −1387.0 | −1325.6 | −1337.1 |
| 15 May 2019 | −1395.3 | −1392.8 | −1343.0 | −1354.0 |
| 1 July 2019 | −1398.7 | −1396.8 | −1358.2 | −1367.1 |
| 5 August 2019 | −1402.0 | −1398.0 | −1368.3 | −1375.2 |
| 5 September 2019 | −1405.0 | −1398.5 | −1375.7 | −1382.9 |
| 15 October 2019 | −1408.0 | −1398.8 | −1383.4 | −1389.5 |
| 15 November 2019 | −1404.3 | −1399.0 | −1389.7 | −1392.7 |
| 15 December 2019 | −1402.2 | −1399.0 | −1392.9 | −1394.7 |
| 15 January 2020 | −1399.3 | −1399.0 | −1394.8 | −1395.9 |
| 15 April 2020 | −1393.9 | −1399.1 | −1396.1 | −1395.5 |
| 25 May 2020 | −1396.3 | −1399.1 | −1395.5 | −1395.6 |
| 15 July 2020 | −1399.3 | −1399.1 | −1395.8 | −1396.5 |

**Table A4.** The subsidence values of the observation point A29 acquired by different methods (mm).

| Phase | Measured Value | Mountainous Time Function Value | EnkF Predicted Value | EnkF Assimilated Value |
|---|---|---|---|---|
| 13 June 2018 | 0.0 | 0.0 | 0.0 | 0.0 |
| 6 July 2018 | −1.3 | −0.2 | −1.0 | −1.3 |
| 3 August 2018 | −11.1 | −9.1 | −10.4 | −10.5 |
| 24 August 2018 | −71.5 | −46.0 | −47.3 | −51.2 |
| 24 October 2018 | −311.1 | −708.5 | −713.6 | −629.0 |
| 15 January 2019 | −1241.1 | −1284.6 | −1205.1 | −1214.4 |
| 15 March 2019 | −1382.4 | −1371.4 | −1301.2 | −1320.8 |
| 15 April 2019 | −1395.9 | −1387.9 | −1337.2 | −1349.6 |
| 15 May 2019 | −1420.2 | −1396.2 | −1358.0 | −1373.5 |
| 1 July 2019 | −1420.4 | −1402.0 | −1379.3 | −1390.4 |
| 5 August 2019 | −1420.6 | −1403.8 | −1392.1 | −1398.6 |
| 5 September 2019 | −1420.7 | −1404.5 | −1399.4 | −1404.6 |
| 15 October 2019 | −1420.8 | −1405.0 | −1405.1 | −1409.3 |
| 15 November 2019 | −1463.0 | −1405.2 | −1409.5 | −1421.5 |

**Table A4.** *Cont*.

| Phase | Measured Value | Mountainous Time Function Value | EnkF Predicted Value | EnkF Assimilated Value |
|---|---|---|---|---|
| 15 December 2019 | −1486.7 | −1405.3 | −1421.7 | −1435.7 |
| 15 January 2020 | −1493.7 | −1405.3 | −1435.5 | −1446.8 |
| 15 April 2020 | −1506.8 | −1405.3 | −1446.9 | −1460.2 |
| 25 May 2020 | −1512.5 | −1405.3 | −1460.2 | −1470.9 |
| 15 July 2020 | −1519.9 | −1405.3 | −1470.8 | −1482.4 |

**Table A5.** The subsidence values of the observation point A36 acquired by different methods (mm).

| Phase | Measured Value | Mountainous Time Function Value | EnkF Predicted Value | EnkF Assimilated Value |
|---|---|---|---|---|
| 13 June 2018 | 0.0 | 0.0 | 0.0 | 0.0 |
| 6 July 2018 | −1.2 | 0.0 | −2.3 | −1.2 |
| 3 August 2018 | −4.6 | −0.3 | −1.5 | −1.8 |
| 24 August 2018 | −8.6 | −1.0 | −2.6 | −3.5 |
| 24 October 2018 | −11.7 | −47.4 | −50.1 | −42.3 |
| 15 January 2019 | −352.0 | −830.1 | −825.1 | −731.7 |
| 15 March 2019 | −1164.1 | −1255.6 | −1157.2 | −1158.8 |
| 15 April 2019 | −1357.6 | −1340.8 | −1244.0 | −1268.7 |
| 15 May 2019 | −1410.4 | −1383.6 | −1311.4 | −1331.7 |
| 1 July 2019 | −1415.6 | −1413.6 | −1361.6 | −1373.2 |
| 5 August 2019 | −1420.6 | −1422.7 | −1382.3 | −1390.4 |
| 5 September 2019 | −1425.3 | −1426.7 | −1394.3 | −1401.5 |
| 15 October 2019 | −1429.8 | −1429.0 | −1403.8 | −1409.2 |
| 15 November 2019 | −1431.0 | −1429.9 | −1410.2 | −1415.0 |
| 15 December 2019 | −1431.6 | −1430.3 | −1415.5 | −1419.4 |
| 15 January 2020 | −1429.1 | −1430.6 | −1419.8 | −1422.0 |
| 15 April 2020 | −1424.6 | −1430.8 | −1422.0 | −1422.6 |
| 25 May 2020 | −1426.6 | −1430.8 | −1422.8 | −1423.6 |
| 15 July 2020 | −1429.1 | −1430.8 | −1423.5 | −1424.6 |

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
