# Peer review of "Ground Subsidence Monitoring in a Mining Area Based on Mountainous Time Function and EnKF Methods Using GPS Data"

_remotesensing, doi:10.3390/rs14246359_

Round 1
Reviewer 1 Report
Dear Authors
Line 55-56 Is GPS a newly developed technology ??? The first GPS satellite was on Earth 44 years ago !!! The GPS system has been in operation for 27 years! Has InSAR also been operating for over 10 years, so what's new?
Line 83 is unnecessary
Line 100 is unnecessary
Fig. 2 - no geographic coordinates
Line 119 is unnecessary
Line 135 is not needed
A large number of patterns. Are formulas 9 and 14 necessary? They can be supplemented with material.
Why is the data from the table repeated on the chart. Leave either the table or the graph. You can put them in the complementary material.
Lack of information about the escarpments under construction - it has an influence on the impact on the pace of the estate.
There is a lack of information on how the method of measurement is selected, i.e. whether to generate data on data and field measurements.
In the introduction, it is worth describing that the mining hollows are used as a tourist attraction of Poland: eg Rurek et al. 2022 "Land".
Introduction - no clear-cut purpose of the research.
Line 78 is covered in section 4, not section 5 ...
Missing from the time of the research - exemplary sinkholes are repeated.
Section 3.1
What was the separation of observation points? Are you relocated to the site?
Dig. 5 - 8 vertical was to have a place that would make a contribution. On the horizontal axis of the set real axis time by date, not pointwise (e.g. between 2018/8/3 / and 2018/8/24 is the same distance as between 2018/10/24 and 2019/1/15). To distort the picture of the chart.
Kind regards
Reviewer 2 Report
Review of the paper “Ground subsidence monitoring and prediction in mountainous mining area by using GPS data and different methods”.
General thoughts
In this paper, the authors present a recurrence equation based on the mountainous time function method and achieve the EnKF method in ground subsidence monitoring and prediction. The research presents not only an effective method for ground subsidence monitoring in mountainous mining areas but provides theoretical support for safe coal mining and ecological environment protection. The prediction results by the two methods and the assimilation results by the EnKF method are approximate to the measured values by GPS monitoring on the whole. At some phases, there are evident deviations for some typical observation points. The large deviations are resulting from the abrupt change during these phases, and the predicted and assimilated values are difficult to adapt to this abrupt change immediately. In addition, the deviations differ largely for various observation points.
Text is written in a logical and thoughtful way, creating a coherent whole, in accordance with the writing regime of a scientific paper (IMRaD). However, some remarks should be rethought. They are described below.
Detailed comments
The list of references, although very long, has numerous shortcomings. The core of the article is Knothe function, whereas the great majority of references are Asian. It looks a little peculiar and arises the question “didn't Knothe have successors?”. No doubt he did. Especially in Poland where he is very respectively treated, as a pioneer and founder of the “school of ground subsidence” (Wikipedia says that he was a member of the Polish Academy of Science, quite a big thing). The short research will show very rich literature, in Polish, Russian, French, German and English. I will skip the old works, e.g. Witold Budryk (some polish sources call “Knothe-Budryk function”, whereas others “Knothe function”) or by Zygmunt Kowalczyk, and focus on the recent articles.
1) Piotr Polanin, “Application of two parameter groups of the Knothe–Budryk theory in subsidence prediction”, Journal of Sustainable Mining, vol. 14(2), 2015, pp.67-75. https://doi.org/10.1016/j.jsm.2015.08.010
2) Guzy A. and Witkowski W.T., "Land Subsidence Estimation for Aquifer Drainage Induced by Underground Mining", Energies,14, 2021, no. 15: 4658. https://doi.org/10.3390/en14154658
3) Witkowski W.T. et al.,. Estimation of Mining-Induced Horizontal Strain Tensor of Land Surface Applying InSAR", Minerals, 11, 2021, no. 7: 788. https://doi.org/10.3390/min11070788
4) Krawczyk A. and Grzybek R., “An evaluation of processing InSAR Sentinel-1A/B data for correlation of mining subsidence with mining induced tremors in the Upper Silesian Coal Basin (Poland)“, E3S Web Conf., no. 26 (2018), p.00003. https://doi.org/10.1051/e3sconf/20182600003
5) Krawczyk A. and Owsianka P., “Using the Bentley MicroStation environment to program calculations of predicted ground subsidence caused by underground mining exploitation”, E3S Web Conf., no. 106 (2019), p. 01003. https://doi.org/10.1051/e3sconf/201910601003
6) Borowski W. and Zyga J., “Geodetic evaluation of terrain surface subsidence around mine shifts of KWK Bogdanka in 1976-1983”, Budownictwo i Architektura, 12(3), 2013, pp. 075-082. https://doi.org/10.35784/bud-arch.1992.
The suggested literature is related to the topic. (2) (3) (4) is the use of InSAR (4th is focused on tremors), (5) is about using Knothe in CAD application, (1) (6) the evaluation of geodetic monitoring. Most of the works are from AGH UST in Krakow (Knothe university). I tried to focus on young researchers, so late works of Knothe are abandoned.
The above-mentioned works are just suggestions. I strongly recommend adding some literature, but what exactly, it is up to the authors. The authors have also a free hand to cite or skip it. If they find something more suitable, then they can cite it as well. Also, the authors might add a couple of additional positions regarding using satellite methods for various applications in mountaineering areas (7-9). More diverse literature will benefit the article.
7) Muntean A. et. al, “A GPS study of land subsidence in the Petrosani (Romania) coal mining area”, Nat. Hazards., vol. 80, 2016, pp. 797–810. https://doi.org/10.1007/s11069-015-1997-y
8) Peyret M. et. al, “Monitoring of the large slow Kahrod landslide in Alborz mountain range (Iran) by GPS and SAR interferometry”, Eng. Geol., vol. 100, 2008, pp. 131–141. https://doi.org/10.1016/j.enggeo.2008.02.013
9) Chwedczuk K. et. al, “Challenges related to the determination of altitudes of mountain peaks presented on cartographic sources”, Geod. Vestn., vol. 66, 2022, pp. 49–59. https://doi.org/10.15292/geodetski-vestnik.2022.01.49-59
There are a few editing and formatting errors, the authors need to check once again, very carefully the whole work. Moreover, the text is not prepared according to the journal’s instructions for authors, e.g. incorrect font, missed spaces e.g. [18].W, etc.
What was the criterion for the selection of Tunlan Minefield instead of others? Is there any specific reason or availability of the data?
Missed line scale on each figure, especially Fig 1 and Fig 2 to ‘feel’ the range of the analysed field by the reader. In my opinion Table 2 and Table 3 for a better interpretation needs to be converted into a graph. There is a lack of description of Figure 3, which shows a big number of data and it is difficult to interpret which subsidence curve is an outlier to another.
After improvements, the text might be accepted for publication.
Reviewer 3 Report
The purpose of this paper is to consider the influence of mountainous in mining area, and authors used mountainous time function, EnKF and mtf-EnKF method to predict the subsidence in typical points. And authors also compared with GPS result. The main comments as follows:
Major comments:
(1)Methods: The time function method considering mountain area and the EnKF method considering mountain area time function method should be the innovation of this paper. However, the author 's description of these two methods was not clear, only the calculation formula of the method is introduced, but the calculation process was not introduced in detail.
(2)The article only uses five monitoring points. From the prediction results of each monitoring point, some prediction error of individual years was very large. How to ensure the accuracy of the method is puzzling.
(3)In the prediction, the article only predicts the value of some monitoring periods, and the significance of the general prediction is to explore the future settlement. What is the significance of this method?
Minor comments:
Line 183-198: These sentences maybe the result part.
Line 205-209: The English description of the parameters was wrong.
Line 212: Where are equations [19]?
Line 212: What is the basis for equation7?
Line 213-221: How to use slope, land use and other parameters in this paper?
Line 449: These sentences maybe the method part.
Line 460-465: This part lacks quantitative description.
Round 2
Reviewer 2 Report
Dear Authors
In general, I considered your work is quality and worth publishing. My comments were suggestions to improve quite good a piece of work. It seems, that you have followed all the revisions. Now, I do not have an objection to suggesting "accept submission". Thank you for the opportunity of reading your work.
Best regards
Reviewer 3 Report
Can be accept